# Understanding the Stress Distribution on Anatomic Customized Root-Analog Dental Implant at Bone-Implant Interface for Different Bone Densities

**DOI:** 10.3390/ma15186379

**Published:** 2022-09-14

**Authors:** Pawhat Nimmawitt, Abdul Azeez Abdu Aliyu, Boonrat Lohwongwatana, Sirida Arunjaroensuk, Chedtha Puncreobutr, Nikos Mattheos, Atiphan Pimkhaokham

**Affiliations:** 1Department of Oral and Maxillofacial Surgery, Faculty of Dentistry, Chulalongkorn University, Bangkok 10330, Thailand; 2Biomedical Engineering Research Center, Chulalongkorn University, Bangkok 10330, Thailand; 3Department of Metallurgical Engineering, Faculty of Engineering, Chulalongkorn University, Bangkok 10330, Thailand; 4Department of Dental Medicine, Karolinska Institute, SE-171 77 Stockholm, Sweden

**Keywords:** customized dental implant, root-analog dental implant, finite element analysis, bone quality, bone density, von mises stress

## Abstract

The aim of this study is to assess the stress distribution on the bone tissue and bone-implant interface of a customized anatomic root-analog dental implant (RAI) by means of finite element analysis (FEA) for different types of bone density. A mandibular right second premolar was selected from the CBCT database. A DICOM file was converted to an STL file to create a CAD model in FEA software. The bone boundary model was created, while bone density types I–IV were determined. Von Mises stress was measured at bone tissues and bone-implant interfaces. To validate the models, the RAI was 3D printed through a laser powder-bed fusion (L-PBF) approach. The results revealed that all RAI designs could not cause plastic deformation or fracture resulting in lower stress than the ultimate tensile stress of natural bone and implant. Compared to a conventional screw-type implant, RAIs possess a more favorable stress distribution pattern around the bone tissue and the bone-implant interface. The presence of a porous structure was found to reduce the stress at cancellous bone in type IV bone density.

## 1. Introduction

For decades dental implants have been improving the quality of life for millions of patients with high predictability, thanks to research and development of implant design and surgical/restorative procedures and devices [1]. Currently, commercially available and most commonly-used dental implants are cylindrical or tapered screw-type, threaded along their length [2]. However, prefabricated implants of these designs cannot easily meet the requirements of the individual oral conditions [3,4]. Customized dental implants, where dimensions and design will perfectly suit the individual patient’s anatomy could address the above mentioned limitations of prefabricated conventional dental implants [5].

The root-analog dental implant (RAI) is a novel customized anatomic dental implant designed to replace one or more tooth roots immediately after extraction. Unlike common titanium screw-type dental implants, RAI is designed to precisely match the extraction socket of a specific patient. This way, RAI could simplify the surgical procedures related to implant placement or even render them obsolete. Furthermore, it could simplify restorative procedures and reduce rehabilitation time, while potentially influencing the dimensional changes of the alveolar ridge [6,7,8]. Other advantages of RAI over conventional screw-type implants include ease of placement with simple tools within minutes, minimal invasiveness, and potential for immediate function [9]. Due to these advantages, RAI can reduce risk associated with implant surgery, patient anatomic landmarks, and operator’s factors [10,11]. Nevertheless, further understanding of the bio-mechanical characteristics of such implant designs remains a prerequisite for further clinical development, evaluation, and success [12,13].

Determining the best bio-mechanical behaviour under different clinical conditions depends on various factors influencing the stress transmitted to the bone-implant interface and the bone tissue. De Faria et al. [12] investigated the effect of prosthetic connections on bone stress around tilted implants and demonstrated that an external hexagon has higher stress concentration on bone tissue than a conical connection. Other factors that have significant influence on the stresses around bone tissues and the bone-implant interface include crown-to-implant ratio [14], designs of implant-supported prostheses [15], implant diameter [16], and bone quality [17,18]. Biomechanically, the stress transmitted to the bone tissue through the bone-implant interface could influence both osseointegration as well as long-term clinical outcomes [19,20]. A thorough understanding of the way in which stress is transmitted to the bone tissue can assist in improving the implant design. While compressive stress to a certain extent can enhance osseointegration at the bone-implant interface, excessive stress will result in bone resorption [20].

Several studies have utilized finite element analysis (FEA) to investigate the stress distribution to the bone tissue with conventional dental implant designs and placement protocols [21,22]. FEA is regarded as an effective tool in assessing stress distribution at the bone-implant interface and thus in investigating how such stress is being transmitted to the bone tissue [23,24]. FEA can simulate bio-mechanical conditions by creating detailed two and three-dimensional models of multiple elements, composed of triangular or quadrilateral areas. Stress distribution for different surfaces and designs of screw-type endosseous implants has been extensively studied [25,26]. There are, however, very limited data with RAI, especially for different types of bone density.

The aim of this study is to assess the stress distribution on the bone-implant interface of RAI for various types of bone density through FEA. The stress distribution for three different types of bone density was assessed with 3 RAI designs and compared to that of a conventional screw-type dental implant.

## 2. Materials and Methods

### 2.1. Design of Custom Digital 3D Models

A mandibular right premolar was retrieved from the radiographic 3D database of Chulalongkorn University, scanned by a 3D Accuitomo 170 machine (J. Morita Inc., Kyoto, Japan). The settings were 5 mA, 90 kV, and 0.25 mm voxel size. The dimensions of the selected premolar (bucco-lingual width = 7.3 mm, mesio-distal width = 5.0 mm, root length = 14.7 mm) fit the average root dimensions according to Woelfel et al. [27].

A digital imaging and communications in medicine (DICOM) file was converted to a standard tessellation language (STL) file using Avizo software 2019.2 (Thermo Fisher Scientific, Waltham, MA, USA) in order to create a 3D tooth model. The STL file was imported to 3-dimensional modelling and finite element software, ANSYS Release 2020 R1 software (ANSYS Inc, Irvine, CA, USA) and converted to a CAD geometry file. The coronal part above the crestal bone was cut off and the remaining part was named design “A” (Figure 1a); thus the final size of the 3D dental model was bucco-lingual width = 6.93 mm, mesio-distal width = 4.48 mm, and root length was 12.25 mm.

Design “B” was created with the 4 most coronal mm identical to design A, but the surface below that point and all the way to the apex was replaced with a porous structure [28] of 2 mm in depth with pore size 600 µm [29]. This resulted in a solid “core” 2 mm below the implant surface extending up to approximately 4 mm coronal of the apex (Figure 1b). Design “C” (Figure 1c) was created by using the 2 most coronal mm of design A, thereafter replacing the rest of the implant model with a porous structure and without the presence of a solid core.

### 2.2. Digital Stock Implant Model

A stock dental implant size 5 × 13 mm was scanned with micro-computed tomography; the settings were 114 µA, 70 kVp, and 20 µm isotropic voxel size at high resolution (1024 × 1024) (µ35 SCANCO Medical, SCANCO Medical AG, Brüttisellen, Switzerland) and converted into a CAD geometry file and named design “D” (Figure 1d).

### 2.3. Bone Boundary Model

The bone boundary model was designed in rectangular form, similar to Inglam et al. [30]. The model’s size was 23.31 mm in height, 10.5 mm bucco-lingual width according to original CBCT-selected data and 5 mm mesio-distal bone away from the dental implant shoulder [31]. Then, four types of bone density according to the Lekholm and Zarb classification [32] were simulated. Type I bone was designed to contain cortical bone only, type II bone contained 2 mm of cortical bone and cancellous bone with normal density, type III bone contained 1 mm of cortical bone and cancellous bone with normal density, and type IV bone contained 1 mm of cortical bone and cancellous bone with low density.

### 2.4. Finite Element Analysis

The FEA model was created based on the material properties recommended by Tada et al. [33] and Santiago et al. [34] shown in Table 1. The implant was assumed to be fully osseointegrated. All elements of the bone model were filled into spaces in the porous structure. The bone-to-implant contact of designs A, B, C, and D were 152.94 mm^2^, 269.32 mm^2^, 324.80 mm^2^, and 198.38 mm^2^, respectively. All materials were considered isotropic, homogeneous, and linear elastic. The constraint definitions were established as fixed in the x, y, and z axis at the mesial and distal boundary surfaces of the cortical and trabecular bone [35]. The vertical force of 200 N and horizontal force of 40 N (from buccal to lingual) were applied to the abutment simultaneously [36]. Von Mises stress was calculated at the cortical bone, cancellous bone, and implant.

### 2.5. Validation of Novel 3D Printed RAI

The design B implant was selected for validation. The RAI was fabricated using the L-PBF process (Mlab cusing 200R; Concept Laser GmbH, Lichtenfels, Germany). Then, the tooth socket model was drilled into a 50 pounds per cubic foot polyurethane foam block, which is equal to type I bone quality (Sawbones^®^; Pacific Research Laboratories Inc., Washington, DC, USA) using a computer numerical control machine (CNC, GENOS M46OR-VE; Okuma Co., Aichi, Japan). The implant was then placed by tapping. The model containing the dental implant was scanned with microCT µ35 SCANCO Medical (SCANCO Medical AG, Brüttisellen, Switzerland) to evaluate the position and printing accuracy of the implant via Avizo software 2020.2 (Thermo Fisher Scientific, Waltham, MA, USA).

## 3. Results

A total of 16 combinations of four types of bone boundary model and four types of implant designs were analyzed by means of FEA as shown in Figure 2a–c.

### 3.1. Von Mises Stress on Cortical Bone

Types II, III and IV bone implant designs were found to have highest stress at the shoulder of the implant (Figure 3a). The lowest stress was found in design A for all the types of bone density. Designs B and D demonstrated similar stress values in bone types I–III, whereas type IV bone density shows a higher stress value. The highest stress value was found in design C. Nevertheless, all cortical bone models produced stress which was less than the ultimate stress tolerance of cortical bone, which is estimated at around 100 MPa [37].

### 3.2. Von Mises Stress on Cancellous Bone

The values of von Mises stress of all models were below the ultimate strength of cancellous bone, (reportedly 8 MPa [37]) except design C in type III bone quality (Figure 3b). In cancellous bone of type II and type III bone quality, the lowest stress was found in the design A implant. However, cancellous bone type IV bone quality had lowest stress in the design B implant. Subsequently, design D showed higher stress than designs A and C.

### 3.3. Von Mises Stress on Implant Models

In type I bone quality, the lowest von Mises stress value was observed in design A (Figure 3c), while higher values were found in design B, design C, and design D. Type II–IV bone quality also revealed design A to have the lowest von Mises stress. However, the highest value was found in design C. All implant models produced stress which was less than the ultimate stress tolerance of the titanium implant, reportedly 900 MPa [37].

### 3.4. Implant Validation

The design B implant was 3D printed (Figure 4a) and placed into a type I bone block socket (Figure 4b). The radiographic image obtained from microCT revealed a minimal vertical discrepancy between the implant and bone block (Figure 5). The vertical discrepancy was measured between the implant surface and bone block using 3D image analysis via Avizo software 2020.2 a 0.05 to 0.10 mm (Figure 5).

The root mean square error (RMSE) was used to evaluate the CAD file compared with the 3D printed implant. A total of 344,016 elements showed that the root mean square (RMS) average was found to be 0.09 mm. Most of the elements (236,817 elements) showed an RMS value between 0 to 0.10 mm. A total of 93,362 elements showed an RMS value between 0.10 to 0.20 mm, while 11,323 elements were found to be between 0.20 to 0.30 mm. Moreover, 2423 elements were between 0.30 to 0.40, and the remaining 91 elements occurred between 0.40 mm to 0.50 mm (Figure 6a). Most of the outer surface showed small RMS values (0 to 0.10), while higher RMS was found in the deeper surfaces of the implant (Figure 6b).

## 4. Discussion

This study investigated the stress distribution at the bone-implant interface of a novel customized anatomic RAI and compared the stress for three different RAI configurations with those of conventional screw-type dental implants for different types of bone density. The results revealed that the RAI design results in lower von Mises stress at both the bone-implant interface and the selected bone types. Our finding is in agreement with that of Chen et al., 2017 [38]. Although an increase in pores density in the designed RAI led to a rise in the von Mises stress in both bone-implant interface and the bone types, design B still showed less stress than design D (control). However, design C exhibited much different stress from the other designs, possibly due to presence of a porous structure without a solid “core”, which may lead to excessive stress concentration at the edge of the struts [39,40]. Stress values within the range of 8–50 MPa might cause plastic deformation of cancellous bone and fracture [37]. Therefore, with regards to stress application, designs A and B can be considered safer than designs C and D. On the other hand, the equivalent stress of 1.6 MPa was reported to adequately prevent the crestal bone loss in the canine mandibular premolar region [41,42], thus preventing disuse bone atrophy.

According to Palka et al. [43] and Barba et al. [29] porous structure with a pore size between 100–600 µm could promote osseointegration. In addition, a porous network could increase the mechanical interlocking between the implant and the bone by allowing the tissues to grow into the porous structure. Shaoki et al. [44] compared the pores volumetric fraction of the dental implant fabricated through L-PBF and other implants’ surface modification approaches. The results revealed no difference in all groups. A porous structure, however, must be below the alveolar bone crest [45]. Brentel et al. [46] analyzed the new bone formation between rough and porous surfaces of the dental implant. The histomorphometric analysis showed that the porous surface was found to have significant bone formation and larger contact surface when compared with the rough surface. Thus, a customized design of a porous implant on the patient’s alveolar ridge anatomy might be beneficial, as opposed to a fully dense implant structure.

Implant surface roughness can be described at the micro or macro level. The porous structure described in this study corresponds to macro level features, while the micro-roughness is not addressed by FEA studies. Micro-roughness is typically the result of chemical or mechanical preparation of the titanium surface with sandblasting or acid-etching and can be assessed with the Sa value corresponding to the difference between the “peak” and the “valley” of the rough elements. According to Albrektsson and Wennerberg [45], implants can be divided into four different categories, depending on the surface roughness measured by the value of Sa: smooth (Sa < 0.5 µm); minimally rough (Sa between 0.5–1.0 µm), moderately rough (Sa between 1.0–2.0 µm); rough (Sa > 2.0 µm). Implants with moderate roughness are reported to have high osseointegration and are the main commercially available surfaces today [47]. L-PBF can generate a rough surface with values between 4–14 µm, depending on machine settings [3,48]. Nevertheless, rough surfaces with Sa more than 2 µm have not shown clinical advantage and may deteriorate problems of peri-implantitis, so have been discontinued.

Bone density can have a strong influence on the stress distribution of screw-type dental implants. Some studies have revealed that thick cortical bone or highly dense cancellous bone lower the level of von Mises stress [21,35]. These results are in agreement with our findings, whereby both screw-type and RAI show the highest von Mises stress in bone type IV around the dental implant. However, the results of this study indicated that the stress is being reduced in cancellous bone in the case of the porous design as described above.

The RMSE is one of the most used methods for calculating a model’s discrepancies in predicting quantitative data [49]. According to model validation, an RMSE value between 0.05–0.1 is acceptable for model manufacturing [50]. Design B was selected because it is the most complex model in this study, while it can still be easily fabricated through the L-PBF process. As seen in Figure 6b, the surface of contact with the surrounding bone presented with less error than any other surface (blue color). Consequently, this model would have a better fit when the implant is placed with tapping of the osteotomy.

A macroscopic surface feature such as porosity is an interesting parameter of the design, which could have clinical implications. Porosity could increase the available surface for osseointegration, while also enhancing the mechanical interlocking of the implant in bone through the ingrowth of bone in the porous structures. Lee et al. [28] evaluated newly mineralized bone and the microarchitecture of trabecular bone that grows in porous dental implants, compared with taper conventional dental implants. Histomorphometry showed faster and more abundant interfacial bone deposition, new bone formation, and remodeling activities in the structurally designed porous dental implants. Although surface behavior for conventional screw-type dental implants has been extensively studied, there is still little understanding of the surface implications with regards to RAIs. Porosity and other macro design elements can, in clinical application, co-exist with any of the commercially available surface modifications at microlevel, which, however, cannot be assessed by FEA.

The results of this study should be seen under the limitations of finite element analysis [51]. Although the value of FEA in simulating mechanical components is undisputed, there are limitations when attempting to simulate living tissues, such as bone. Calculation of the initial stress distribution on the bone, cannot predict tissue outcomes as it cannot estimate the bone response. The laboratory validation also cannot simulate bone ingrowth in a porous structure. In addition, masticatory force can vary widely among patients, as well as the force direction and duration patterns, especially in patients with parafunctions [52]. Nevertheless, by comparing different structures it can offer indications of potential clinical implications when selecting each of the four different designs. Full tight contact between the implant and the surrounding bone was assumed, although in practice osseointegration has been shown in histology to occur with 50–70% bone to implant contact. Nevertheless, the impact of the percentage of bone to implant contact on the stress distribution is suggested to be minimal and could not alter results of FEA [53]. Furthermore, two types of stress have been mostly used in scientific literature to evaluate dental implant and surrounding bone: von Mises stress and maximum principal stress. Von Mises stress was selected in this study, as this approach can evaluate overall stress in materials such as bone, which possess both ductile and brittle qualities [54]. Furthermore, bone is an anisotropic material, but the assumption was made in this study to analyze it as isotropic in order to simplify the calculations. The bone models were generated as simply as possible to simulate general conditions of homogenous and isotropic bone, which might differ in diverse anatomic positions such as the ramus. Finally, the stock implant selected for design D corresponds to a common clinical choice of implant and its size is similar, but not identical, to the premolar root and the RAIs. The second premolar was selected in this study, because it had the least variation compared to other teeth [55].

## 5. Conclusions

The stress distribution at the bone-implant interface and the bone tissue of RAI was investigated through FEA. The following specific conclusions can be drawn:The RAI was found to have a more favorable stress distribution pattern at the bone tissues and the bone-implant interface than the conventional screw-type implant in all types of bone density.The presence of a porous structure reduced the stress in cancellous bone of type IV.All designs demonstrated lower stress than what could cause plastic deformation or fracture based on the ultimate stress tolerance of both the bone and implant.Type I bone presents the lowest stress values at both bone tissue and the bone-implant interface than the other bone types.

## Figures and Tables

**Figure 1 materials-15-06379-f001:**
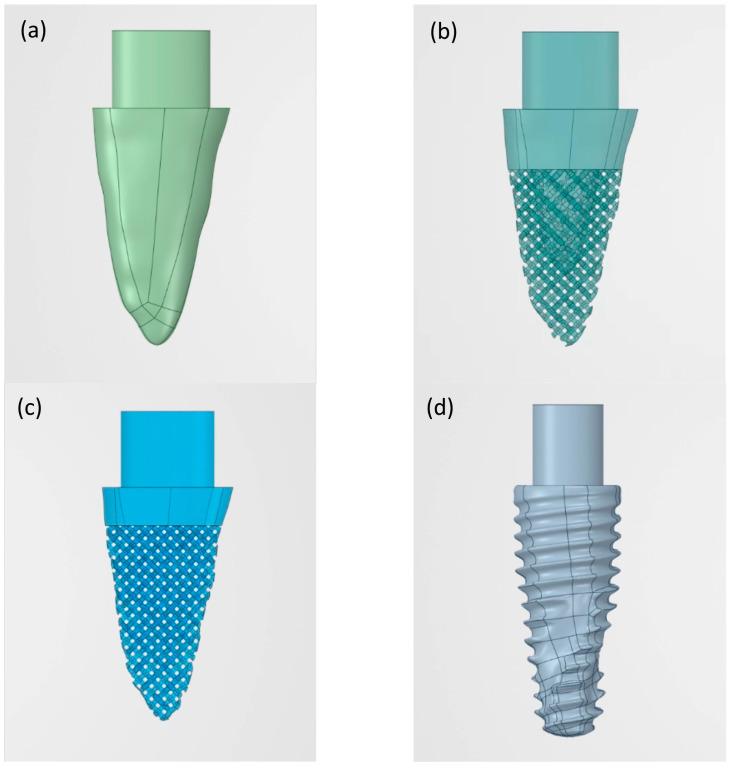
CAD models of four implant designs: (**a**) Solid root-analog implant; (**b**) Solid and porous root-analog implant; (**c**) Porous root-analog implant; (**d**) Screw-type implant (control).

**Figure 2 materials-15-06379-f002:**
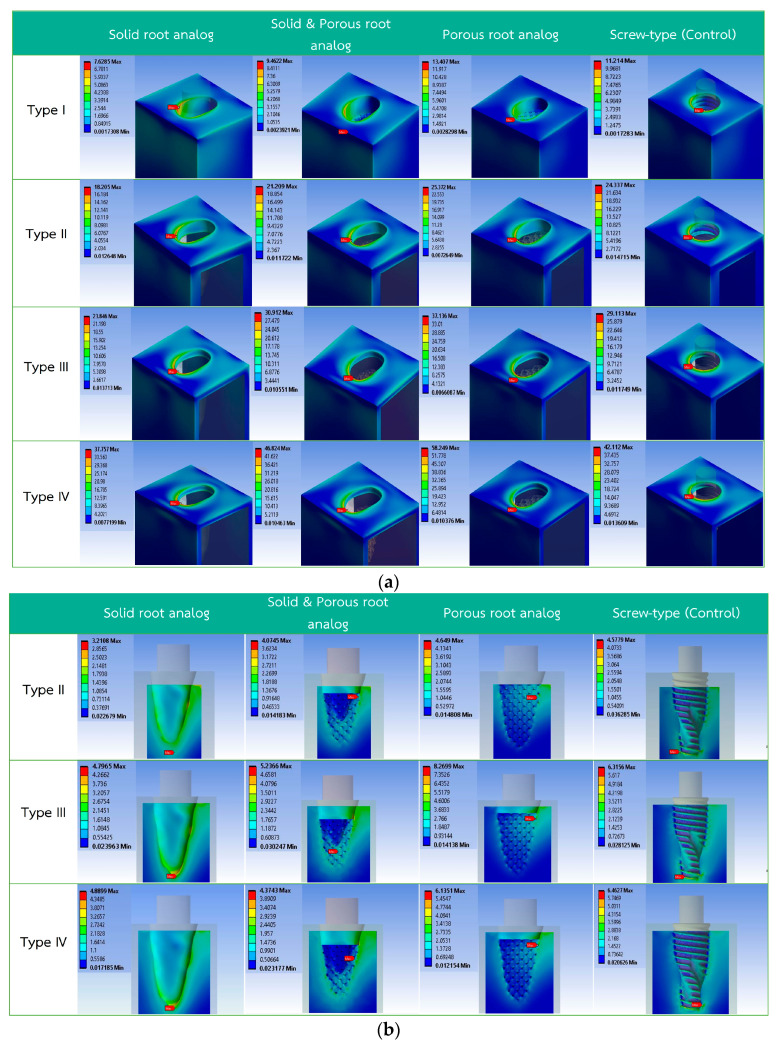
Von Mises stress distribution of all finite element models: (**a**) von Mises stress in cortical bone; (**b**) von Mises stress in cancellous bone; (**c**) von Mises stress in implant.

**Figure 3 materials-15-06379-f003:**
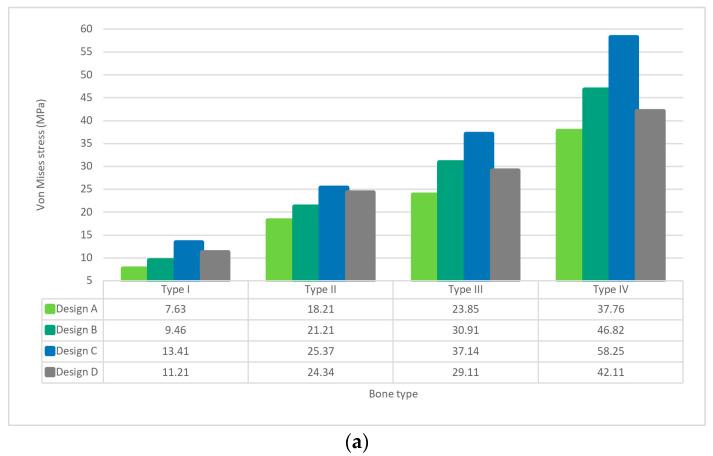
Bar chart of von Mises stress in all models; (**a**) von Mises stress in cortical bone; (**b**)von Mises stress in cancellous bone; (**c**) von Mises stress in implant.

**Figure 4 materials-15-06379-f004:**
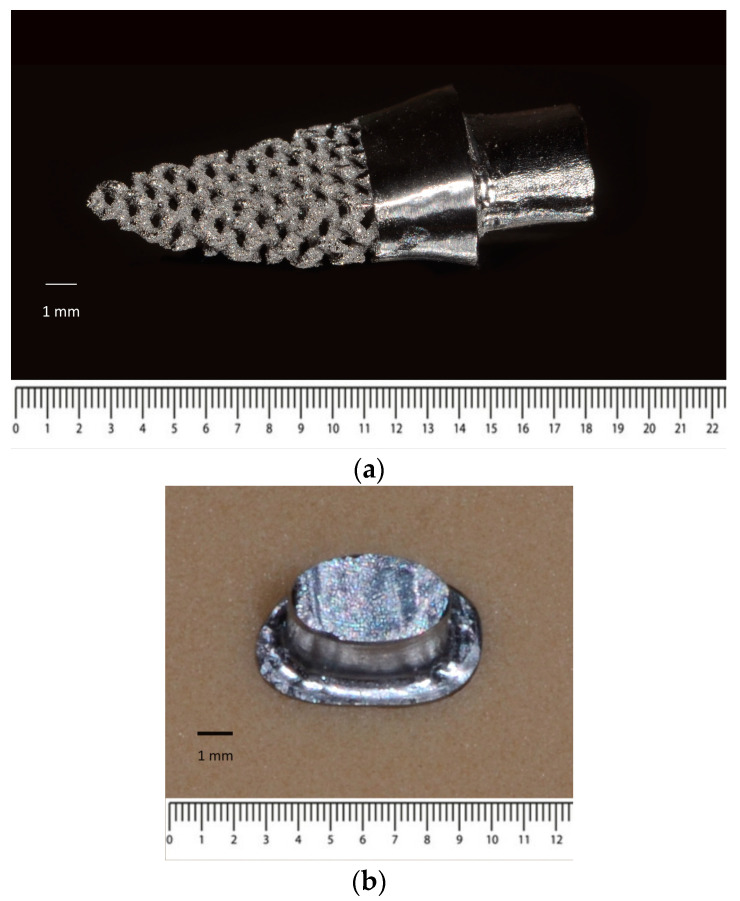
Solid and porous RAI, fabricated through L-PBF technique (**a**) RAI (**b**) implant placed in bone block.

**Figure 5 materials-15-06379-f005:**
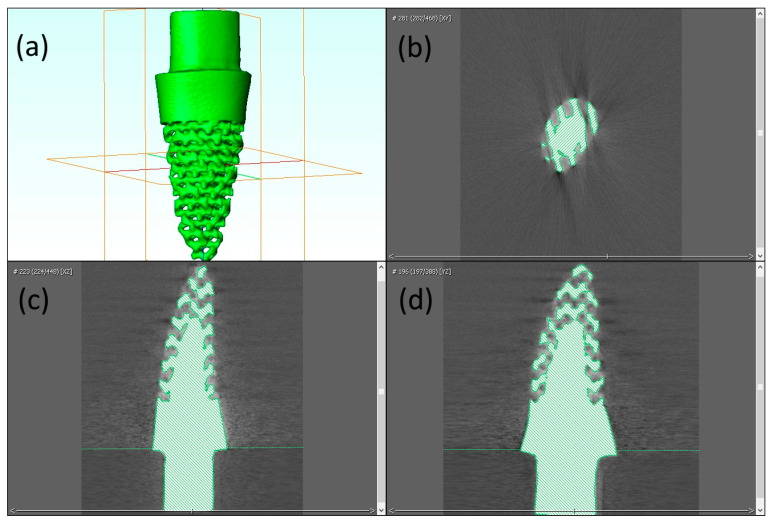
Radiographic image of design B implant placed in bone block: (**a**) 3D view; (**b**)axial view; (**c**) coronal view; (**d**) sagittal view.

**Figure 6 materials-15-06379-f006:**
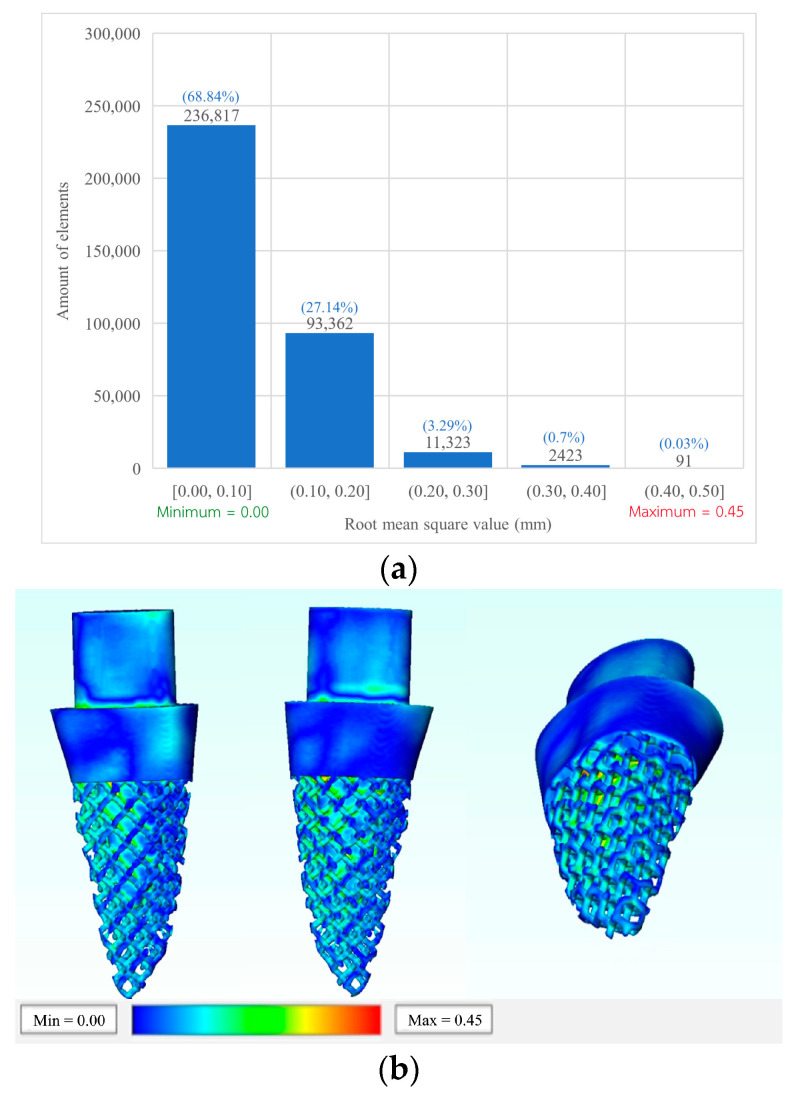
Root mean square value of the implant model (**a**) bar chart of elements represent in root mean square value; (**b**) 3D model of root mean square.

**Table 1 materials-15-06379-t001:** Characteristics of the FEA models of various types of bone and titanium implant.

Materials	Modulus of Elasticity (GPa)	Poisson’s Ratio (V)
Cortical bone	13.7	0.3
Cancellous bone (bone type II, II)	1.37	0.3
Cancellous bone (bone type IV)	0.69	0.3
Titanium	110.0	0.35

## Data Availability

The data sets used and/or analyzed during the current study are available from the corresponding author upon reasonable request.

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
