# Peer review of "Understanding the Stress Distribution on Anatomic Customized Root-Analog Dental Implant at Bone-Implant Interface for Different Bone Densities"

_materials, 2022, doi:10.3390/ma15186379_

Round 1

Reviewer 1 Report

Manuscript ID: materials-1889985

Type: Article

Title: 
Understanding the stress distribution on anatomic customized root-analog dental implant at bone-implant interface for different bone densities

English form: English language and style are fine/minor spell check required

Abstract: reformulate in a better way this section (specify what is RAI...)

Introduction: "Determining the best bio-mechanical behaviour under different clinical conditions 49 depends on various factors influencing the stresses transmitted to the bone-implant inter- 50 face and the bone tissue [12, 13]. Biomechanically, the stress transmitted on the bone tis- 51 sues through the bone-implant interface could influence both osseointegration as well 52 long term clinical outcomes [14] [15] [16]. A thorough understanding of the way through 53which the stress is transmitted to the bone tissue can assist in improving the implant de- 54 sign. While compressive stress of certain extend can enhance osseointegration at bone- 55 implant interface, excessive stress will result in bone resorption [15]"
Explain which parameters could affect the stresses transmitted to the bone-implant interface and the bone tissue including references (Examples: too insertion torque, bone density, temperature, surgical implant procedure) [PMID: 34348016 - doi.org/10.3390/app10238623 - PMID: 29328609] 

Methods: very good.

Discussion:Explain in a better way the study results. 

Evidence the study limitations and explain why the clinical scenario could appear very different from FEA analysis.

After correction, manuscript must to be revaluate.

Reviewer 2 Report

In this article, the author reported the stress distribution on the RAI implant interface. This is informative for many study in the field and benefits the future study. From the perspective of academic criticism, several technical concerns need to be addressed to further improve the quality of this manuscript, as appended below.

1. The rationale of the CAD design should be explained more to cover the clinical relevance if there is any.

2. The legends in Figure 2 are very hard to read. Please change the font size and optimize the image resolution accordingly.

3. In Figure 3, why is the Von mises stress so different from design to design in material b and c but not much in a? How does the design affect the mechanical properties in this case? Please extend the discussion to cover these questions.

4. The surface roughness of the implant significantly contributes to the success of implantation. The different designs mentioned here seem to result in very different surface roughness. Is it a critical factor here?

Round 2

Reviewer 1 Report

Authors improved manuscript following reviewers suggestion.

I recommend English form review and the checking of references according to the Mendely format.

Reviewer 2 Report

Accept in present form